# Microstructure and Mechanical Properties of AA1050/AA6061 Laminated Composites Fabricated through Three-Cycle Accumulative Roll Bonding and Subsequent Cryorolling

**DOI:** 10.3390/ma17030577

**Published:** 2024-01-25

**Authors:** Lingling Song, Haitao Gao, Zhengyu Wang, Huijie Cui, Charlie Kong, Hailiang Yu

**Affiliations:** 1Light Alloys Research Institute, Central South University, Changsha 410083, China; songlingling163@163.com (L.S.); gaohaitao@csu.edu.cn (H.G.); 2Shimadzu (China) Co., Ltd., Shanghai 200233, China; sshwzy@shimadzu.com.cn (Z.W.); cuihuijie1984@163.com (H.C.); 3Mark Wainwright Analytical Centre, University of New South Wales, Sydney, NSW 2052, Australia; c.kong@unsw.edu.au

**Keywords:** AA1050/AA6061 laminated composites, accumulative roll bonding, cryorolling, microstructure, mechanical properties

## Abstract

In this study, AA1050/AA6061 laminated composites were prepared by three-cycle accumulative roll bonding (ARB) and subsequent rolling. The effects of the rolling process on the microstructure evolution and mechanical properties of AA1050/AA6061 laminated composites were systematically investigated. The results indicate that the mechanical properties of the laminated composites can be effectively improved by cryorolling compared with room-temperature rolling. The microstructure analysis reveals that cryorolling can suppress the necking of the hard layer to obtain a flat lamellar structure. Moreover, the microstructure characterized by transmission electron microscopy shows that cryorolling can inhibit the dynamic recovery and significantly refine the grain size of the constituent layers. Meanwhile, the tensile fracture surface illustrates that AA1050/AA6061 laminated composites have the optimal interfacial bonding quality after cryorolling. Therefore, the laminated composites obtain excellent mechanical properties with the contribution of these factors.

## 1. Introduction

With the rapid development of technology and industry, the diversified demand for material properties is increasing. Composites have been extensively considered for their excellent properties. Among them, laminated metal composites (LMCs) composed of two or more metals are widely used in aerospace, automotive, mechanical electronics, petrochemical, and other fields due to their outstanding fracture toughness, fatigue life, corrosion resistance, and damping ability [1,2,3,4]. The selection of component metals is crucial for the final performance of LMCs. Similarly, factors such as interlayer strain coordination, interface structural state, thickness ratio of component metals, and microstructure evolution can also have a great impact on the comprehensive properties of LMCs [5,6,7]. In contrast to most LMCs consisting of dissimilar metals, aluminum laminated composites do not introduce unnecessary influencing factors such as intermetallic compounds at the interface [8]. Moreover, aluminum alloy has the advantages of high specific strength, good electrical and thermal conductivity, light weight, corrosion resistance, and low price, making it one of the most commonly used metals [9,10]. In practical production, aluminum laminated composites have a wider application prospect owing to their preparation process being simpler compared to that of other LMCs, ensuring high interfacial bonding strength [11].

Many methods have been used to fabricate LMCs, such as explosive bonding, extrusion bonding, diffusion bonding, rolling bonding, and so on [12,13,14,15]. The accumulative roll bonding (ARB) process in rolling bonding is considered a promising method for preparing aluminum laminated composites because of its high production efficiency, simple equipment operation, and wide application range [16,17]. The ARB process has obvious advantages in the preparation of LMCs. It can break the limitation of the conventional rolling deformation reduction ratio, introduce large strain into the component metals, and obtain good plate shape with almost no change in geometry. At the same time, with the increase in strain, the grain size can be significantly refined and the strength of the material can be improved. Degner et al. [18] produced AA1050/AA7075 laminated composites through ARB and further improved their formability via short-term heat treatment. The results showed that these aluminum laminated composites not only had sufficient strength but also had good formability and corrosion resistance, which can be used to process structural components such as automotive body panels. Yuan et al. [19] prepared AA6082/AA7204 laminated composites using ARB and analyzed the relationship between grain orientation and corrosion behavior. The study found that, compared with those with R cube texture and brass texture, the grains near S texture had poor corrosion resistance. However, with the increase in the rolling reduction ratio, more grains in the AA7204 layer near the bonding interface rotated into brass texture, improving their corrosion resistance. Kümmel et al. [20] fabricated AA2024/AA5005 laminated composites by ARB and studied their fatigue life. The results indicated that, compared with single materials with coarse and ultrafine grains, the fatigue life of AA2024/AA5005 laminated composites was significantly improved. Although the ARB process can obviously refine the grain size and improve the mechanical properties of aluminum laminated composites, the enhancement effect is relatively limited due to plastic instability. Therefore, the preparation of LMCs through various rolling processes has become a current research hotspot, for example, using cryorolling as a subsequent deformation treatment method. The low-temperature condition in the cryorolling process is maintained by liquid nitrogen, which can suppress dynamic recovery and achieve higher accumulated dislocation density. These dislocations will promote the formation of a large number of nucleation positions, thereby forming subgrain or ultrafine grain materials. However, research reports on further strengthening aluminum laminated composites by combining cryorolling are relatively finite. Compared with other processes for preparing LMCs, we not only use ARB in the severe plastic deformation methods to obtain aluminum laminated composites but also improve their comprehensive properties through subsequent cryorolling, mainly utilizing the characteristics of cryorolling to suppress dynamic recovery and overcome limited grain refinement.

In the present investigation, the AA1050/AA6061 laminated composites were fabricated by the ARB process. The effects of subsequent room-temperature rolling and cryorolling on the microstructure and mechanical properties of AA1050/AA6061 laminated composites were studied. The mechanism of cryorolling in enhancing mechanical properties was elaborated, and its influence on the interfacial bonding quality and microstructure evolution was analyzed.

## 2. Materials and Methods

The experimental materials used in this work were annealed AA1050 and solution-treated AA6061 (provided by Guangdong Hongwang New Materials Technology Co., Ltd., Shenzhen, China) with sizes of 230 × 75 × 1 mm^3^. The AA1050/AA6061 laminated composites were processed through ARB and subsequent rolling. The corresponding preparation process is shown in Figure 1. Firstly, the eight-layer laminated composites were prepared by three-cycle ARB. The main purposes were to ensure the coordinated deformation between the component metals as much as possible, restrain the plastic instability of the hard layer, and obtain AA1050/AA6061 laminated composites with a flat lamellar structure and excellent properties. The detailed process flow was as follows: The acetone and alcohol were used to clear greasy dirt and impurities attached to the surface of the materials, and then the oxide film was removed using wire brushing. The sheets were stacked and then fixed with iron wire at the corners to avoid relative slip during rolling. Subsequently, the first-cycle ARB was performed at room temperature, resulting in a 50% reduction in the thickness of the laminated composites. The purpose of conducting ARB at room temperature is to improve the tensile strength of the laminated composite as much as possible while ensuring good deformation coordination. The obtained AA1050/AA6061 laminated composites were then cut in half, cleaned, and stacked again, and the rolling continued at a 50% reduction in thickness. This preparation process was repeated three times to acquire eight-layer laminated composites. Secondly, after three-cycle ARB, the AA1050/AA6061 laminated composites were subjected to room-temperature rolling (RTR) and cryorolling (CR), respectively, with a final thickness of 0.25 mm. Samples to be cryorolled were soaked in liquid nitrogen for 30 min before rolling. The laminated composites prepared by different rolling processes are defined as A1 sample (one-cycle ARB), A2 sample (two-cycle ARB), A3 sample (three-cycle ARB), A3 + RTR sample (three-cycle ARB + RTR), A3 + CR sample (three-cycle ARB + CR). 

The tensile properties of AA1050/AA6061 laminated composites were determined through a Shimadzu AGS-X 10-kN tensile machine (Shimadzu, Kyoto, Japan) with a tensile rate of 1 × 10^−3^ s^−1^. Dog-bone-shaped tensile samples were machined to have a parallel length of 13 mm and a width of 2.5 mm along the rolling direction. The microhardness was measured by the Vicker’s hardness tester HXD-2000TMC/LCD 181101X (Shanghai Taiming Optical Instrument Co., Ltd., Shanghai, China) with a load of 100 g and a holding time of 15 s. The lamellar structure distribution of the rolling direction–normal direction cross-section was observed by optical microscopy (OM, Mshot MJ42, Mingmei Optoelectronic Technology Co., Ltd., Guangzhou, China). Field Emission Gun Transmission Electron Microscopy (FEG-TEM, Philips Electron Optics, Eindhoven, The Netherlands) was adopted to analyze the microstructure evolution of the AA1050/AA6061 matrix and bonding interface, and the equipment operated at 200 kV. The tensile fracture morphologies of the initial and rolled samples were examined through scanning electron microscopy (SEM, TESCAN MIRA3 LMU, Zeiss Sigma 300, Shanghai, China).

## 3. Results and Discussion

### 3.1. Mechanical Properties of AA1050/AA6061 Laminated Composites

Figure 2 shows the trend of microhardness changes in the AA1050 and AA6061 layers under different states. It can be observed from Figure 2a that the Vicker’s hardness value of the initial annealed-state AA1050 was 26.6 ± 0.2 HV, and the Vicker’s hardness value of AA6061 after solution treatment was 56.6 ± 1.0 HV. The right side of the dashed line in the figure shows the microhardness of the two constituent layers of the A3, A3 + RTR, and A3 + CR samples. Compared with the initial state, the microhardness values of the AA1050 and AA6061 layers showed a significant increasing trend after severe plastic deformation. Many studies have shown that microhardness is mainly influenced by several factors [21], such as: (1) the generation of dislocations in the material and its work hardening behavior, (2) the size, distribution, and quantity of particles in the matrix, such as the second phases and precipitates, (3) the rearrangement and annihilation of dislocations (work softening), etc. The first two factors increase the microhardness, while the latter factor leads to a decrease in the microhardness. For the AA1050/AA6061 laminated composites, the microhardness increased with the increase in rolling deformation, but the AA6061 layer had a faster growth rate and a more obvious change in hardness value, as shown in Figure 2b. According to the performance characteristics of the material, the difference in the microhardness growth rate between the two constituent layers is closely related to its cold working rate.

In order to further understand the microhardness changes of AA1050/AA6061 laminated composites after ARB, Vicker’s hardness testing was carried out in the layer thickness direction. The experimental results are shown in Figure 3. It can be seen that the microhardness of the AA6061 layer was always higher than that of the AA1050 layer, and, as the ARB process continued, the microhardness in the layer thickness direction increased gradually. This can indicate that the hardness of both component metals increased due to work hardening caused by rolling. Similarly, the growth amplitude of the microhardness of the AA6061 layer was greater than that of the AA1050 layer in the layer thickness direction, which is related to the higher strain hardening rate of the AA6061 layer. By observing Figure 3, it can be found that there is a significant jump in the hardness values at the bonding interface of the first three-cycle ARB, but they still remain between the two constituent layers. A similar conclusion can be drawn according to Table 1. For the A1 sample, the average hardness at the interface was 65.7 ± 2.0 HV, while that of the constituent layers AA1050 and AA6061 was 44.6 ± 0.8 HV and 107.1 ± 3.8 HV, respectively. Then, the average hardness gradually increased with the continuation of ARB. When it came to the A3 sample, this increased to 71.2 ± 1.5 HV, 48.1 ± 1.0 HV, and 122.6 ± 1.5 HV, respectively. Some studies suggest that the change in microhardness at the interface of the LMCs can indicate that the constituent layers are directly bonded during the rolling process without forming intermetallic compounds [22]. The increase in microhardness throughout the plastic deformation process is mainly attributed to work hardening, the accumulation of dislocation, and the dispersion distribution of the second-phase particles in the AA6061 layer [23].

Figure 4 illustrates the tensile properties of the initial materials and the rolled AA1050/AA6061 laminated composites. The true stress–strain curves of annealed AA1050 and solution-treated AA6061, as well as A3 + RTR and A3 + CR samples, are shown in Figure 4a. The results indicate that, after severe plastic deformation, the true stress of the laminated composites significantly increased while the true strain decreased, which is related to the work hardening generated during the deformation process. At the same time, it can be found that the tensile properties of the A3 + CR sample were better than those of the A3 + RTR sample, which means that, compared with RTR, CR can obviously enhance the performance of AA1050/AA6061 laminated composites. The variation in the ultimate tensile strength (UTS) and elongation of the samples under different deformation states are shown in Figure 4b. It can be seen from the figure that the UTS continuously increased with the increase in deformation amount and reached 236 ± 1 MPa after three-cycle ARB. Then, the RTR and CR were carried out, and the UTS of the samples was 298 ± 2 MPa and 310 ± 1 MPa, respectively, which increased by 26% and 31% compared to three-cycle ARB. The elongation decreased throughout the entire plastic processing, which is consistent with other reports on the preparation of LMCs using severe plastic deformation processes [24,25].

Figure 5 displays the tensile properties of AA1050/AA6061 laminated composites under two rolling processes and their variations with deformation states. According to the changes in the UTS and elongation after three-cycle ARB and RTR in Figure 5a,b, it can be seen that the growth ratio of the UTS decreased with the increase in ARB cycle (16.81%→10.17%). After RTR, the growth ratio of the UTS showed an upward trend, which is due to the shift in the rolling process from severe plastic deformation to conventional plastic processing, resulting in a change in the performance trend of the laminated composites. The variation of elongation in Figure 5b is also similar, but, contrary to the UTS growth ratio, the elongation continuously decreased with the increase in deformation, and the decline ratio became faster after shifting the rolling process. This is related to the strain hardening effect and dislocation density accumulation during plastic deformation [26]. The tensile properties and their change regulation of the laminated composites after three-cycle ARB and CR are shown in Figure 5c,d. It can be found that the overall trend of the UTS and elongation was similar to that under RTR. However, the analysis results showed that the UTS growth ratio of the laminated composites was higher after CR, while the elongation decline ratio was lower. This demonstrates that, compared with RTR, the laminated composites after CR had better mechanical properties, which can be attributed to the characteristics of CR. During this deformation process, the dynamic recovery of the component metals was inhibited, resulting in the accumulation of higher dislocation density [27], so that the UTS was significantly improved.

### 3.2. Microstructure Analysis of AA1050/AA6061 Laminated Composites 

Figure 6 shows the lamellar structure morphologies of AA1050/AA6061 laminated composites after rolling deformation. The structural evolution of each constituent layer of the laminated composites under different rolling states was examined by OM. Based on the experimental results and combined with the different deformation capabilities of the component metals during rolling processing, it can be inferred that the brighter layer was AA6061, while the corresponding layer was AA1050. From Figure 6a to Figure 6b, it can be observed that the two component metals can maintain relatively uniform deformation in the first three-cycle ARB. The lamellar structure showed a flat morphology. Moreover, no obvious pores or defects were found at the bonding interface, indicating that the AA1050/AA6061 interface introduced in each cycle had achieved a relatively good bonding. However, as the rolling deformation continued, the local necking phenomenon occurred in the hard layer (AA6061 layer) of the laminated composites after RTR and CR, and some lamellar structures exhibited curved morphologies (as shown in Figure 6c,d). According to the literature research, due to the differences in mechanical properties of each component metal, it is difficult for LMCs to coordinate deformation during rolling, which can easily lead to plastic instability of the hard layer [28]. However, in this study, the AA6061 layer did not fracture under large deformation, and the overall lamellar structure was relatively stable. Therefore, the strengthening effect of subsequent rolling on the laminated composites was still significant, which was reflected in the continuous improvement of the UTS in tensile properties, and the UTS of the A3 + CR sample was higher. Many studies have shown that CR can make the accumulated dislocation density reach a higher steady-state level by inhibiting the dynamic recovery, and these dislocations will serve as the driving force to initiate a large number of nucleation sites, ultimately achieving the effect of refining grains and improving the tensile strength of materials [29,30]. By observing and analyzing the evolution of the lamellar structure of AA1050/AA6061 laminated composites under different rolling states, it can be found that there is no obvious plastic instability, and the constituent layers always maintain a continuous distribution, which has a good corresponding relationship with the changes in their mechanical properties.

The microstructure of A3, A3 + RTR, and A3 + CR samples characterized by TEM is shown in Figure 7. Obviously, the laminated composites under different rolling states exhibited sharp bonding interfaces. Meanwhile, the bonding interface of A3 + RTR and A3 + CR samples is compared and analyzed. It is observed that the interface morphology can be improved by CR to obtain the laminated composites with the flat interface (as shown in Figure 7d,g). In addition, Figure 7 also shows the microstructure evolution of the AA1050 and AA6061 layers. The AA6061 layer of the A3 sample included many refined, elongated grains containing high-density dislocation, while the AA1050 layer still displayed relatively coarse grains with low-density dislocation, as shown in Figure 7b,c. As the deformation continued, the grain size of the two constituent layers decreased after RTR and CR, and the grain refinement effect of the AA1050 layer in the cryorolled sample is more evident than that of the RTR sample in the comparison of Figure 7e,h. Although the introduction of more high-energy grain boundaries through grain refinement is effective in hindering dislocation movement and improving strength, their ability to accommodate plastic deformation is also compromised by reduced ductility and toughness [31]. Therefore, the AA1050/AA6061 laminated composites prepared by combining multiple rolling processes (ARB, RTR, and CR) revealed excellent UTS and lower elongation (as shown in the tensile properties in Figure 4 and Figure 5). Among them, the UTS of the cryorolled sample increased most significantly, which is mainly related to improving the interface morphology, inhibiting dynamic recovery, accumulating high dislocation density, and further refining grain size.

Figure 8 illustrates the tensile fracture morphologies of the initial samples, including annealed AA1050 and solution-treated AA6061. The significant necking of AA1050 during the tensile process can be observed from Figure 8a, indicating that the material had excellent ductility. The morphology of the dimples (large and deep dimples) in Figure 8b also confirms this view. Compared with AA1050, the elongation of AA6061 after solution treatment had decreased, which was due to the presence of more alloying elements in AA6061. Based on Figure 8c,d, no obvious necking can be found before its tensile fracture, and the dimples on the fracture surface became smaller and shallower, which can explain the lower elongation compared to AA1050. There are studies indicating that aluminum alloys have two macroscopic fracture modes, namely, shear fracture mode and dimple fracture mode [32,33]. From the perspective of the microscopic mechanism, shear fracture is caused by the cracking and aggregation of microscopic shear planes within the material. Ductile fracture is the nucleation, growth, and aggregation of voids induced by inclusions or second-phase particles in the material until they connect with macroscopic cracks, resulting in ductile crack propagation and material failure. According to the characteristics of the dimples in the fracture surface, it can be determined that the failure mechanism of both initial materials is a ductile fracture.

Figure 9 characterizes the fracture morphologies of tensile samples under different rolling states. Figure 9a,b shows the tensile fracture of AA1050/AA6061 laminated composites prepared by one-cycle ARB at low and high magnification. The delamination phenomenon can be clearly observed from the fracture surface, indicating that the component metals were not well bonded during initial rolling. This is because of the significant difference in hardness between the AA1050 and AA6061 layers, which makes it difficult to bond them together during coordinated deformation, and they are prone to debonding after the tensile test. In addition, the AA1050 layer exhibited a more evident necking than the AA6061 layer, which means that the AA1050 layer contributed more to the elongation of the laminated composites during the deformation process. It can be found that the size and number of dimples decreased visibly compared to the initial materials at high magnification, showing that the ductility of the A1 sample was greatly reduced, which was consistent with the tensile test results. The fracture surface of the A3 sample is shown in Figure 9c,d, and the interface delamination can still be observed in the sample. Especially for the interface introduced by the third-cycle ARB, its bonding strength was relatively weak, so the delamination phenomenon was also quite obvious. Comparing the tensile fracture surface of the laminated composites fabricated by the RTR and CR (Figure 9e–h), some weak bonding positions at the fracture surface can be noticed after RTR. However, the interfacial bonding quality had been greatly improved through CR, and no delamination occurred on the fracture surface. This means that the AA1050/AA6061 laminated composites produced by CR have excellent mechanical properties, which can be confirmed by the tensile test results in Figure 4. According to the fracture morphologies of the AA1050 and AA6061 layers under different rolling states, most of the fracture surfaces had a gray-fiber appearance with hemispherical dimples, which was the result of the formation and aggregation of micropores. It is worth mentioning that the dimples at the fracture surface of some LMCs that have undergone ARB exhibit morphological characteristics of being stretched in a specific direction, which is considered unique to shear dimples. Research has shown that the generation of shear dimples is mainly due to two reasons [34]. Firstly, the shear stress inside the material during tensile testing can lead to the formation of micropores and stretching in specific directions. The second reason is that severe and non-uniform deformation during the ARB process results in the occurrence of shear bands in the material. Under the action of stress, the metal flows in the direction of shear bands, causing the micropores inside the material to be “pulled” and eventually forming shear dimples.

The characterization and analysis of the microstructure of AA1050/AA6061 laminated composites mentioned above indicate that interfacial bonding quality is one of the critical indicators for evaluating comprehensive properties. Many studies have shown that the interface of LMCs plays an important role in the deformation process, mainly reflected in load transfer, regulation, and redistribution of stress and strain [35]. Strong interface bonding is the prerequisite for LMCs to obtain outstanding mechanical properties. Liu et al. [36] reported that good interface bonding can delay local necking and premature fracture of the hard layer in component metals, improve the distribution of the constituent layers, and thus enhance the ability for uniform plastic deformation. Chen et al. [37] found that Al/Ti laminated composites with a strong interface coupling effect achieved extraordinary elongation which was significantly higher than that of the single material. Therefore, it is meaningful to study the bonding mechanism of LMCs fabricated by the rolling process. In recent years, numerous studies have reported several recognized bonding mechanisms, including recrystallization theory, energy theory, diffusion theory, thin-film theory, N. Bay theory, and so on. It is generally believed that there are two bonding modes at the interface during the rolling deformation process, namely, the fresh metal bonding zone and residual thin layer refining zone [38]. Figure 10 shows the schematic illustration of interface evolution. The thin oxide film and work hardening layer on the treated surfaces have much lower ductility than the matrix, making them prone to breaking during plastic deformation. The fresh metals exposed on both sides will be extruded under the rolling pressure, and then the atoms of the two fresh metals reach the atomic level distance at the fracture position, attracting each other. Thereby, the metallurgical bonding forms at the interface. However, there are still some areas where thin films are mixed between the constituent layers, and their hardness is greater than that of the matrix. Therefore, as the deformation amount further increases, more obvious grain refinement will occur in these areas, forming the interface bonding differently from the former.

## 4. Conclusions

In this paper, AA1050/AA6061 laminated composites were prepared through ARB and subsequent rolling. The mechanical properties and microstructure evolution of the laminated composites under different rolling states were explored. The main conclusions are drawn as follows:The AA1050/AA6061 laminated composites were successfully processed by three-cycle ARB with the UTS of 236 MPa. After subsequent rolling, the UTS of A3 + RTR and A3 + CR samples increased to 298 MPa and 310 MPa, respectively. Therefore, the mechanical properties of AA1050/AA6061 laminated composites can be obviously enhanced by CR;Analyzing the evolution of the microstructure, compared with RTR, CR can improve the interface structure morphology, inhibit dynamic recovery, accumulate higher dislocation density, and further refine grain size. These factors highly contribute to the mechanical improvement of AA1050/AA6061 laminated composites;The fracture analysis results reveal that the interfacial delamination existed at the tensile fracture surface of AA1050/AA6061 laminated composites that underwent ARB, and weak bonding positions were also found in the room-temperature rolled sample. In contrast, good bonding between the constituent layers was observed in the fracture morphology of the cryorolled sample, indicating that CR can effectively improve the interfacial bonding quality.

## Figures and Tables

**Figure 1 materials-17-00577-f001:**
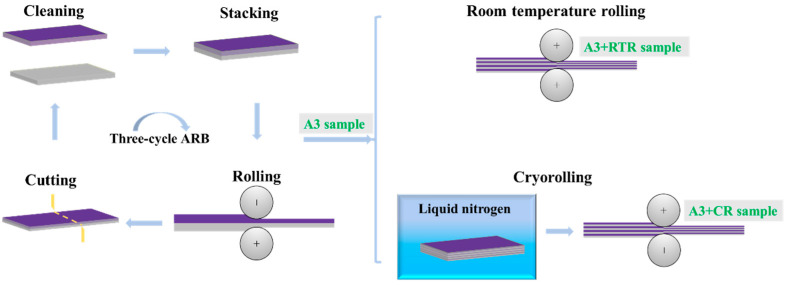
Schematic illustration of ARB and subsequent rolling processes.

**Figure 2 materials-17-00577-f002:**
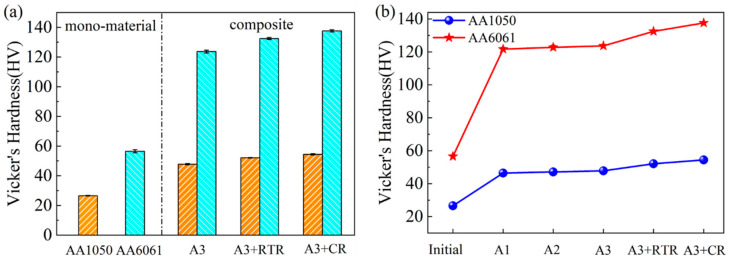
Vicker’s hardness of the initial materials and AA1050/AA6061 laminated composites under different rolling states: (**a**) histogram, (**b**) line chart.

**Figure 3 materials-17-00577-f003:**
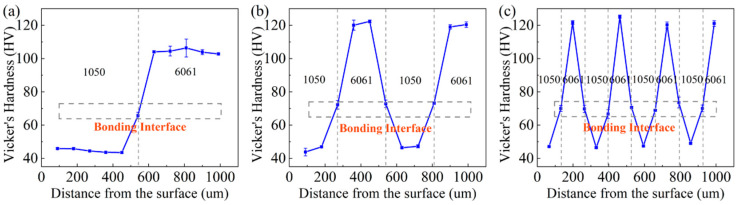
Microhardness distribution in the layer thickness direction of AA1050/AA6061 laminated composites: (**a**) A1 sample, (**b**) A2 sample, (**c**) A3 sample.

**Figure 4 materials-17-00577-f004:**
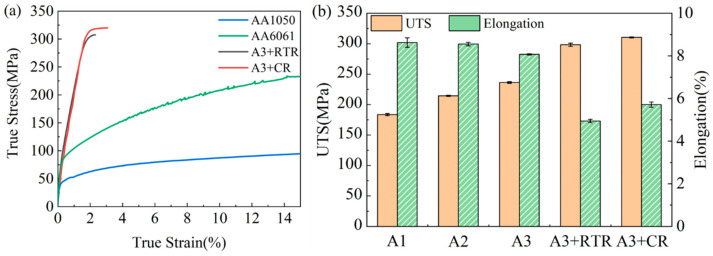
Tensile properties of AA1050/AA6061 laminated composites: (**a**) true stress–strain curves, (**b**) the variation of UTS and elongation with rolling states.

**Figure 5 materials-17-00577-f005:**
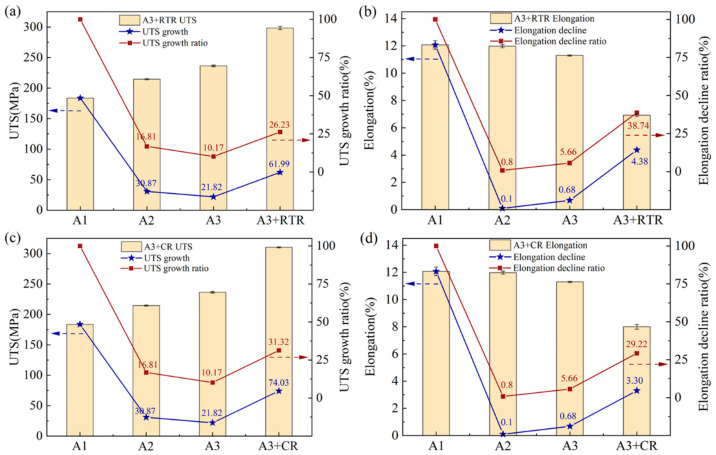
The variation of tensile properties of AA1050/AA6061 laminated composites: (**a**,**b**) A3 + RTR sample, (**c**,**d**) A3 + CR sample.

**Figure 6 materials-17-00577-f006:**
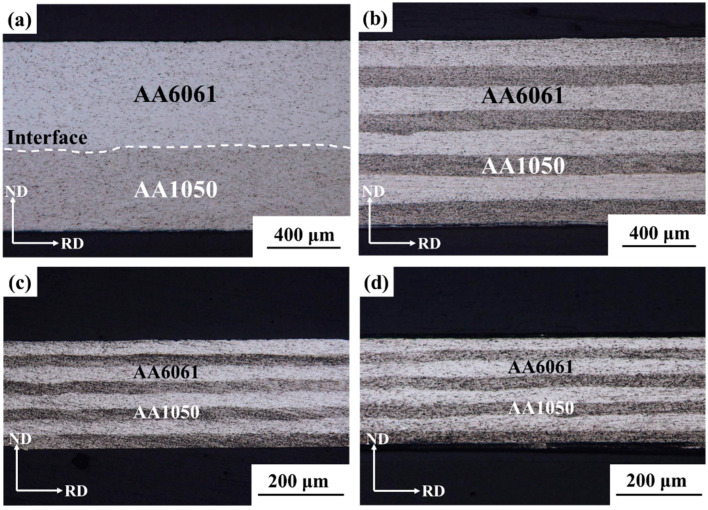
OM images of AA1050/AA6061 laminated composites prepared through ARB and subsequent rolling: (**a**) A1 sample, (**b**) A3 sample, (**c**) A3 + RTR sample, (**d**) A3 + CR sample.

**Figure 7 materials-17-00577-f007:**
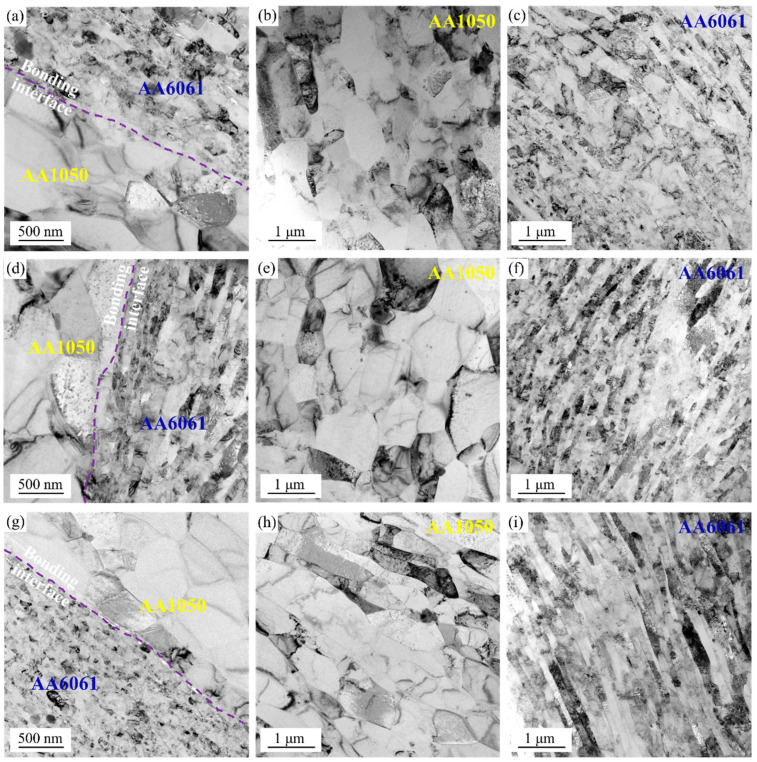
TEM micrographs of AA1050/AA6061 laminated composites under different rolling states: (**a**–**c**) A3 sample, (**d**–**f**) A3 + RTR sample, (**g**–**i**) A3 + CR sample.

**Figure 8 materials-17-00577-f008:**
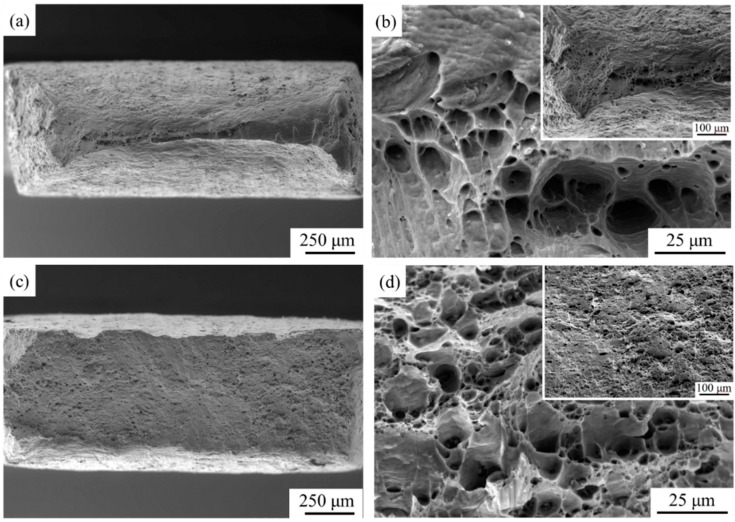
Tensile fracture of the initial materials: (**a**,**b**) AA1050, (**c**,**d**) AA6061.

**Figure 9 materials-17-00577-f009:**
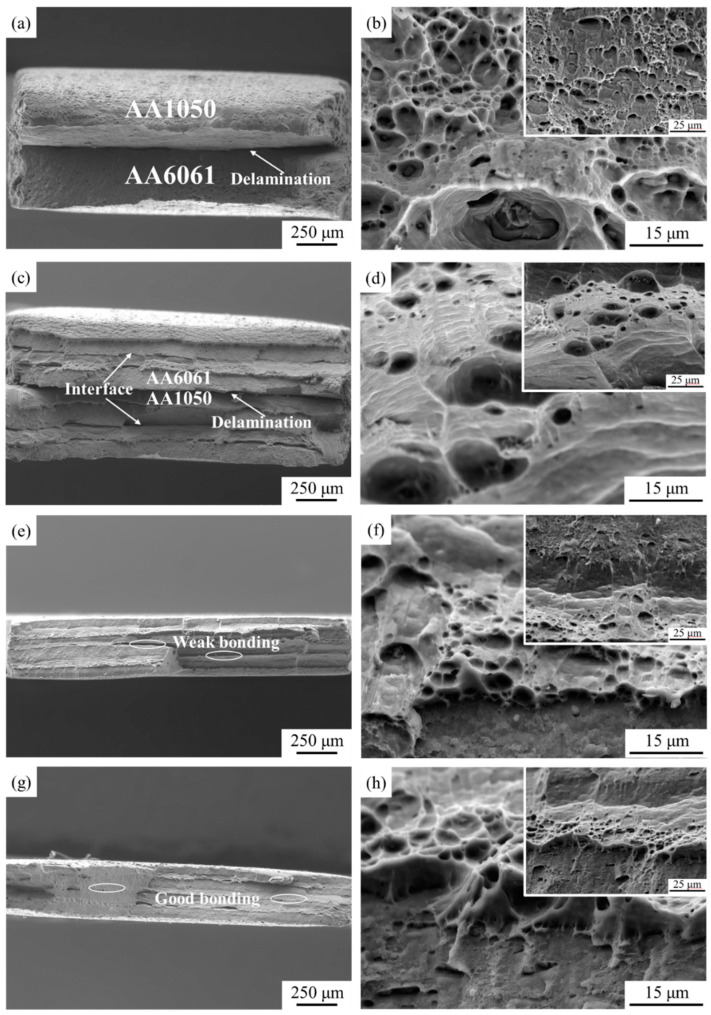
Tensile fracture surface of AA1050/AA6061 laminated composites: (**a**,**b**) A1 sample, (**c**,**d**) A3 sample, (**e**,**f**) A3 + RTR sample, (**g**,**h**) A3 + CR sample.

**Figure 10 materials-17-00577-f010:**
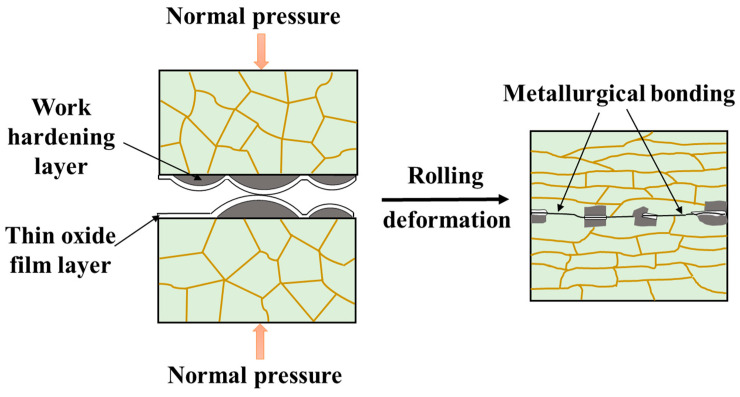
Schematic illustration of rolling bonding mechanism.

**Table 1 materials-17-00577-t001:** Vicker’s hardness (HV) in the layer thickness direction of A1, A2, and A3 samples.

As-Rolled	AA1050	AA1050/AA6061	AA6061
A1	44.64 ± 0.81	65.67 ± 1.96	107.14 ± 3.81
A2	47.43 ± 2.84	70.77 ± 1.05	119.28 ± 2.35
A3	48.05 ± 0.99	71.24 ± 1.51	122.60 ± 1.47

## Data Availability

The data presented in this study are available on request from the corresponding author. The data are not publicly available due to ongoing studies.

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
