# Peer review of "Microstructure and Mechanical Properties of AA1050/AA6061 Laminated Composites Fabricated through Three-Cycle Accumulative Roll Bonding and Subsequent Cryorolling"

_materials, 2024, doi:10.3390/ma17030577_

Round 1

Reviewer 1 Report

Comments and Suggestions for Authors

The paper presents the laminated metal layered composite consisting of two different aluminum alloys that have different compositions and heat treatment history. The metal rolling  was used to join two sheets and finally to obtain a relatively thin sheet of layered metal by rolling on room temperature and at low temperatures.

Some comments to the presentation of results:

In figure 5 it is not very clear what is presented in the diagram, I conclude that it is the increase in properties but I am still not very sure how to read those graphs.

I see that the low temperature enables better lamination and better contact between layers, it is proved by micrographs. Did you analyze the interface in two layered composites in terms of composition variation at the interphase. This could be interesting to read

Author Response

The paper presents the laminated metal layered composite consisting of two different aluminum alloys that have different compositions and heat treatment history. The metal rolling was used to join two sheets and finally to obtain a relatively thin sheet of layered metal by rolling on room temperature and at low temperatures.

Some comments to the presentation of results:

  1. In figure 5 it is not very clear what is presented in the diagram, I conclude that it is the increase in properties but I am still not very sure how to read those graphs.

Answer: Thank you for your kind suggestion. I'm sorry to have caused you confusion. Figure 5 describes the changes in UTS and elongation of AA1050/AA6061 laminated composites prepared by subsequent room temperature rolling and cryorolling. The main purpose is to demonstrate from multiple aspects that the cryorolled samples have better tensile properties. The UTS growth ratio of the sample after cryorolling is higher than that of room temperature rolling, so its UTS value is also the highest, while the elongation decline ratio is lower than that of room temperature rolling, so its deformation capacity is stronger. We have added the following content in the fourth paragraph of the results and discussion section of the manuscript: The main purpose is to compare the mechanical properties of room temperature rolled and cryorolled samples from multiple aspects.

  1. I see that the low temperature enables better lamination and better contact between layers, it is proved by micrographs. Did you analyze the interface in two layered composites in terms of composition variation at the interphase. This could be interesting to read.

Answer: Thank you for your kind suggestion. We analyzed the change and diffusion of elements at the interface of AA1050/AA6061 laminated composites by Energy Dispersive Spectroscopy (EDS), and the results are shown in Figure A. Since the element that can most distinguish the component metals in the two aluminum alloys is the Mg element (with a high content in AA6061), the experimental results focus on the Mg element. It can be observed from the figure that there is a certain degree of element diffusion at the interface under different deformation states.

Figure A. The area scan of Mg elements in AA1050/AA6061 laminated composites through EDS: (a) A3 + RTR sample, (b) A3 + CR sample.

Reviewer 2 Report

Comments and Suggestions for Authors

Comments and specific questions requiring clarification:

1.       Introduction is to short in my opinion and should be expanded. You present general information about aluminium alloys application and some information about research results of using of accumulative roll bonding and cryorolling process. But basic information about theory of accumulative roll bonding and cryorolling process should be added in my opinion. Add some theory of these two processes if it is possible – main idea, maybe some diagram e.t.c.

2.       Explain please – why You decided exactly to three passes during ARB process? What was the reason? If it is possible add this information to the paper please.

3.       As You know mechanical properties are depended  of plastic working process parameters – for example - temperature among other thing. Add information (to the introduction or to the paragraph no. 2) information – why You decided to the cold rolling of ARB process? Is it possible to obtain similar properties by hot ABR rolling process with thermal treatment after the rolling process? It is important question when we want to connect materials with different mechanical properties (big difference in stress for example). Sometimes it can’t be possible in cold ARB rolling process. It can be easier in hot conditions. But then additional thermal treatment after the rolling process is necessary. I know it depends also what kind of properties we want to obtain.

4.       Add please short paragraph to the introduction – what is a novelty of Your research compared to the previously publicated paper – it is necessary in my opinion.

5.       Add please information about force value and loading time during microhardnes measurement.

6.       Please consider modification of the paper title. In my opinion better title will be: Microstructure and Mechanical Properties of AA1050/AA6061  Laminated Composites Fabricated through Accumulative Roll 3 Bonding and Cryorolling.

7.       Add please more conclusions based on Yours research results. Only two conclusions is to little in my opinion.

8.       I recommend not to divide the words. If it is possible please correct it. It applies to the whole manuscript.

Author Response

  1. Introduction is to short in my opinion and should be expanded. You present general

information about aluminium alloys application and some information about research results of using of accumulative roll bonding and cryorolling process. But basic information about theory of accumulative roll bonding and cryorolling process should be added in my opinion. Add some theory of these two processes if it is possible – main idea, maybe some diagram e.t.c.

Answer: Thank you for your kind suggestion, which is very useful for our paper. In the introduction, we add the following basic principles and mechanisms of accumulative roll bonding and cryorolling processes: The ARB process has obvious advantages in the preparation of LMCs. It can break the limitation of conventional rolling deformation reduction ratio, introduce large strain into the component metals, and obtain good plate shape with almost no change in geometry. At the same time, with the increase of strain, the grain size can be significantly refined and the strength of the material can be improved. For example, using cryorolling as a subsequent deformation treatment method. The low-temperature condition in the cryorolling process is maintained by liquid nitrogen, which can suppress dynamic recovery and achieve higher steady-state levels of accumulated dislocation density. These dislocations will serve as the driving force to initiate a large number of nucleation positions, thereby forming subgrain or ultrafine grain materials.

  1. Explain please – why You decided exactly to three passes during ARB process? What was the reason? If it is possible add this information to the paper, please.

Answer: Thank you for your kind suggestion. By preparing LMCs with ARB, it is easy to produce uncoordinated deformation in the rolling process due to the performance differences among the component metals. According to the experimental exploration, the component metals can ensure relatively coordinated deformation in the first three-cycle ARB. The hard layer did not appear plastic instability (Figure. B shows the optical micrograph of the AA1050/AA6061 laminated composite after five-cycle ARB, and it is obvious that the hard layer produces obvious necking and fracture), and the constituent layers can maintain a flat state, which can keep the excellent mechanical properties of aluminum laminated composites. In the manuscript, we have added the following contents in the experimental section: The main purpose is to ensure the coordinated deformation between the component metals as much as possible, restrain the plastic instability of the hard layer, and obtain AA1050/AA6061 laminated composites with a flat lamellar structure and excellent properties.

Figure. B Optical micrograph of AA1050/AA6601 laminated composite fabricated through five-cycle ARB.

  1. As You know mechanical properties are depended of plastic working process parameters – for example – temperature among other thing. Add information (to the introduction or to the paragraph no. 2) information – why You decided to the cold rolling of ARB process? Is it possible to obtain similar properties by hot ABR rolling process with thermal treatment after the rolling process? It is important question when we want to connect materials with different mechanical properties (big difference in stress for example). Sometimes it can’t be possible in cold ARB rolling process. It can be easier in hot conditions. But then additional thermal treatment after the rolling process is necessary. I know it depends also what kind of properties we want to obtain.

Answer: Thank you for your kind suggestion. Firstly, for the laminated composites obtained through rolling, we chose aluminum alloys as the component metals, so the performance differences between the two metals are much smaller than those between the dissimilar metals. Secondly, AA6061 was subjected to solid solution treatment before rolling, making it more prone to plastic deformation and reducing its incompatibility with AA1050. Finally, rolling at room temperature can enable the laminated composites to achieve superior tensile strength. This study mainly explores the performance changes of AA1050/AA6061 laminated composites after plastic deformation and analyzes the advantages and mechanisms of cryorolling in improving mechanical properties. Therefore, no subsequent heat treatment research is carried out. According to your suggestion, we have added the following content to the materials and methods section of the manuscript: The purpose of conducting ARB at room temperature is to improve the tensile strength of the laminated composite as much as possible while ensuring good deformation coordination.

  1. Add please short paragraph to the introduction – what is a novelty of Your research

compared to the previously published paper – it is necessary in my opinion.

Answer: Thank you for your kind suggestion. At the end of the second paragraph of the introduction, we described the limitations of only using the ARB process and proposed the preparation of aluminum laminated composites by combining cryorolling. According to your suggestion, we add the following content in the introduction of the manuscript: Compared with other processes for preparing LMCs, we not only use ARB in the severe plastic deformation methods to obtain aluminum laminated composites but also improve their comprehensive properties through subsequent cryorolling, mainly utilizing the characteristics of cryorolling to suppress dynamic recovery and overcome limited grain refinement.

  1. Add please information about force value and loading time during microhardness

measurement.

Answer: Thank you for your kind suggestion. The description of relevant experiment in the manuscript has been modified as follows: The microhardness was measured by the Vicker's hardness tester HXD-2000TMC/LCD 181101X, with a load of 100 g and a holding time of 15 s.

  1. Please consider modification of the paper title. In my opinion better title will be:

Microstructure and Mechanical Properties of AA1050/AA6061 Laminated Composites

Fabricated through Accumulative Roll 3 Bonding and Cryorolling.

Answer: Thank you for your kind suggestion. We have revised the paper title in the manuscript as follows: Microstructure and Mechanical Properties of AA1050/AA6061 Laminated Composites Fabricated through Three-cycle Accumulative Roll Bonding and Subsequent Cryorolling.

  1. Add please more conclusions based on Yours research results. Only two conclusions is to little in my opinion.

Answer: Thank you for your kind suggestion. We have modified the conclusions of the manuscript as follows: (2) Analyzing the evolution of microstructure, compared with RTR, CR can improve the interface structure morphology, inhibit dynamic recovery, accumulate higher dislocation density, and further refine grain size. These factors highly contribute to the mechanical improvement of AA1050/AA6061 laminated composites. (3) The fracture analysis results reveal that the interfacial delamination existed at the tensile fracture surface of ARBed AA1050/AA6061 laminated composites, and weak bonding positions were also found in the room temperature rolled sample. In contrast, good bonding between the constituent layers was observed in the fracture morphology of the cryorolled sample, indicating that CR can effectively improve the interfacial bonding quality.

  1. I recommend not to divide the words. If it is possible, please correct it. It applies to the whole manuscript.

Answer: Thank you for your kind suggestion. We have corrected the divided words throughout the whole manuscript.
